# A Unified Approach for Learning the Parameters of Sum-Product Networks

**Han Zhao**
Machine Learning Dept.
Carnegie Mellon University
han.zhao@cs.cmu.edu

**Pascal Poupart**
School of Computer Science
University of Waterloo
ppoupart@uwaterloo.ca

**Geoff Gordon**
Machine Learning Dept.
Carnegie Mellon University
ggordon@cs.cmu.edu

## Abstract

We present a unified approach for learning the parameters of Sum-Product networks (SPNs). We prove that any complete and decomposable SPN is equivalent to a mixture of trees where each tree corresponds to a product of univariate distributions. Based on the mixture model perspective, we characterize the objective function when learning SPNs based on the maximum likelihood estimation (MLE) principle and show that the optimization problem can be formulated as a signomial program. We construct two parameter learning algorithms for SPNs by using sequential monomial approximations (SMA) and the concave-convex procedure (CCCP), respectively. The two proposed methods naturally admit multiplicative updates, hence effectively avoiding the projection operation. With the help of the unified framework, we also show that, in the case of SPNs, CCCP leads to the same algorithm as Expectation Maximization (EM) despite the fact that they are different in general.

## 1 Introduction

Sum-product networks (SPNs) are new deep graphical model architectures that admit exact probabilistic inference in linear time in the size of the network [14]. Similar to traditional graphical models, there are two main problems when learning SPNs: structure learning and parameter learning. Parameter learning is interesting even if we know the ground truth structure ahead of time; structure learning depends on parameter learning , so better parameter learning can often lead to better structure learning. Poon and Domingos [14] and Gens and Domingos [6] proposed both generative and discriminative learning algorithms for parameters in SPNs. At a high level, these approaches view SPNs as deep architectures and apply projected gradient descent (PGD) to optimize the data log-likelihood. There are several drawbacks associated with PGD. For example, the projection step in PGD hurts the convergence of the algorithm and it will often lead to solutions on the boundary of the feasible region. Also, PGD contains an additional arbitrary parameter, the projection margin, which can be hard to set well in practice. In [14, 6], the authors also mentioned the possibility of applying EM algorithms to train SPNs by viewing sum nodes in SPNs as hidden variables. They presented an EM update formula without details. However, the update formula for EM given in [14, 6] is incorrect, as first pointed out and corrected by [12].

In this paper we take a different perspective and present a unified framework, which treats [14, 6] as special cases, for learning the parameters of SPNs. We prove that any complete and decomposable SPN is equivalent to a mixture of trees where each tree corresponds to a product of univariate distributions. Based on the mixture model perspective, we can precisely characterize the functional form of the objective function based on the network structure. We show that the optimization problem associated with learning the parameters of SPNs based on the MLE principle can be formulated as a signomial program (SP), where both PGD and exponentiated gradient (EG) can be viewed as first order approximations of the signomial program after suitable transformations of the objective

function. We also show that the signomial program formulation can be equivalently transformed into a difference of convex functions (DCP) formulation, where the objective function of the program can be naturally expressed as a difference of two convex functions. The DCP formulation allows us to develop two efficient optimization algorithms for learning the parameters of SPNs based on sequential monomial approximations (SMA) and the concave-convex procedure (CCCP), respectively. Both proposed approaches naturally admit multiplicative updates, hence effectively deal with the positivity constraints of the optimization. Furthermore, under our unified framework, we also show that CCCP leads to the same algorithm as EM despite that these two approaches are different from each other in general. Although we mainly focus on MLE based parameter learning, the mixture model interpretation of SPN also helps to develop a Bayesian learning method for SPNs [21].

PGD, EG, SMA and CCCP can all be viewed as different levels of convex relaxation of the original SP. Hence the framework also provides an intuitive way to compare all four approaches. We conduct extensive experiments on 20 benchmark data sets to compare the empirical performance of PGD, EG, SMA and CCCP. Experimental results validate our theoretical analysis that CCCP is the best among all 4 approaches, showing that it converges consistently faster and with more stability than the other three methods. Furthermore, we use CCCP to boost the performance of LearnSPN [7], showing that it can achieve results comparable to state-of-the-art structure learning algorithms using SPNs with much smaller network sizes.

## 2 Background

### 2.1 Sum-Product Networks

To simplify the discussion of the main idea of our unified framework, we focus our attention on SPNs over Boolean random variables. However, the framework presented here is general and can be easily extended to other discrete and continuous random variables. We first define the notion of *network polynomial*. We use $\mathbb{I}_x$ to denote an indicator variable that returns 1 when $X = x$ and 0 otherwise.

**Definition 1** (Network Polynomial [4]). Let $f(\cdot) \geq 0$ be an unnormalized probability distribution over a Boolean random vector $\mathbf{X}_{1:N}$. The network polynomial of $f(\cdot)$ is a multilinear function $\sum_{\mathbf{x}} f(\mathbf{x}) \prod_{n=1}^{N} \mathbb{I}_{\mathbf{x}_n}$ of indicator variables, where the summation is over all possible instantiations of the Boolean random vector $\mathbf{X}_{1:N}$.

A Sum-Product Network (SPN) over Boolean variables $\mathbf{X}_{1:N}$ is a rooted DAG that computes the network polynomial over $\mathbf{X}_{1:N}$. The leaves are univariate indicators of Boolean variables and internal nodes are either sum or product. Each sum node computes a weighted sum of its children and each product node computes the product of its children. The *scope* of a node in an SPN is defined as the set of variables that have indicators among the node's descendants. For any node $v$ in an SPN, if $v$ is a terminal node, say, an indicator variable over $X$, then $\text{scope}(v) = \{X\}$, else $\text{scope}(v) = \bigcup_{\tilde{v} \in Ch(v)} \text{scope}(\tilde{v})$. An SPN is *complete* iff each sum node has children with the same scope. An SPN is *decomposable* iff for every product node $v$, $\text{scope}(v_i) \bigcap \text{scope}(v_j) = \varnothing$ where $v_i, v_j \in Ch(v), i \neq j$. The scope of the root node is $\{X_1, \ldots, X_N\}$.

In this paper, we focus on complete and decomposable SPNs. For a complete and decomposable SPN $\mathcal{S}$, each node $v$ in $\mathcal{S}$ defines a network polynomial $f_v(\cdot)$ which corresponds to the sub-SPN (subgraph) rooted at $v$. The network polynomial of $\mathcal{S}$, denoted by $f_{\mathcal{S}}$, is the network polynomial defined by the root of $\mathcal{S}$, which can be computed recursively from its children. The probability distribution induced by an SPN $\mathcal{S}$ is defined as $\text{Pr}_{\mathcal{S}}(\mathbf{x}) \triangleq \frac{f_{\mathcal{S}}(\mathbf{x})}{\sum_{\mathbf{x}} f_{\mathcal{S}}(\mathbf{x})}$. The normalization constant $\sum_{\mathbf{x}} f_{\mathcal{S}}(\mathbf{x})$ can be computed in $O(|\mathcal{S}|)$ in SPNs by setting the values of all the leaf nodes to be 1, i.e., $\sum_{\mathbf{x}} f_{\mathcal{S}}(\mathbf{x}) = f_{\mathcal{S}}(\mathbf{1})$ [14]. This leads to efficient joint/marginal/conditional inference in SPNs.

### 2.2 Signomial Programming (SP)

Before introducing SP, we first introduce geometric programming (GP), which is a strict subclass of SP. A *monomial* is defined as a function $h : \mathbb{R}_{++}^n \mapsto \mathbb{R}: h(\mathbf{x}) = d x_1^{a_1} x_2^{a_2} \cdots x_n^{a_n}$, where the domain is restricted to be the positive orthant ($\mathbb{R}_{++}^n$), the coefficient $d$ is positive and the exponents $a_i \in \mathbb{R}, \forall i$. A *posynomial* is a sum of monomials: $g(\mathbf{x}) = \sum_{k=1}^{K} d_k x_1^{a_{1k}} x_2^{a_{2k}} \cdots x_n^{a_{nk}}$. One of the key properties of posynomials is positivity, which allows us to transform any posynomial into the log

domain. A GP in standard form is defined to be an optimization problem where both the objective function and the inequality constraints are posynomials and the equality constraints are monomials. There is also an implicit constraint that $\mathbf{x} \in \mathbb{R}_{++}^n$.

A GP in its standard form is not a convex program since posynomials are not convex functions in general. However, we can effectively transform it into a convex problem by using the *logarithmic transformation trick* on $\mathbf{x}$, the multiplicative coefficients of each monomial and also each objective/constraint function [3, 1].

An SP has the same form as GP except that the multiplicative constant $d$ inside each monomial is not restricted to be positive, i.e., $d$ can take any real value. Although the difference seems to be small, there is a huge difference between GP and SP from the computational perspective. The negative multiplicative constant in monomials invalidates the logarithmic transformation trick frequently used in GP. As a result, SPs cannot be reduced to convex programs and are believed to be hard to solve in general [1].

## 3 Unified Approach for Learning

In this section we will show that the parameter learning problem of SPNs based on the MLE principle can be formulated as an SP. We will use a sequence of optimal monomial approximations combined with backtracking line search and the concave-convex procedure to tackle the SP. Due to space constraints, we refer interested readers to the supplementary material for all the proof details.

### 3.1 Sum-Product Networks as a Mixture of Trees

We introduce the notion of *induced trees* from SPNs and use it to show that every complete and decomposable SPN can be interpreted as a mixture of induced trees, where each induced tree corresponds to a product of univariate distributions. From this perspective, an SPN can be understood as a huge mixture model where the effective number of components in the mixture is determined by its network structure. The method we describe here is not the first method for interpreting an SPN (or the related arithmetic circuit) as a mixture distribution [20, 5, 2]; but, the new method can result in an exponentially smaller mixture, see the end of this section for more details.

**Definition 2** (Induced SPN). Given a complete and decomposable SPN $\mathcal{S}$ over $X_{1:N}$, let $\mathcal{T} = (\mathcal{T}_V, \mathcal{T}_E)$ be a subgraph of $\mathcal{S}$. $\mathcal{T}$ is called an *induced SPN* from $\mathcal{S}$ if

1. $Root(\mathcal{S}) \in \mathcal{T}_V$.
2. If $v \in \mathcal{T}_V$ is a sum node, then exactly one child of $v$ in $\mathcal{S}$ is in $\mathcal{T}_V$, and the corresponding edge is in $\mathcal{T}_E$.
3. If $v \in \mathcal{T}_V$ is a product node, then all the children of $v$ in $\mathcal{S}$ are in $\mathcal{T}_V$, and the corresponding edges are in $\mathcal{T}_E$.

**Theorem 1.** If $\mathcal{T}$ is an induced SPN from a complete and decomposable SPN $\mathcal{S}$, then $\mathcal{T}$ is a tree that is complete and decomposable.

As a result of Thm. 1, we will use the terms *induced SPNs* and *induced trees* interchangeably. With some abuse of notation, we use $\mathcal{T}(\mathbf{x})$ to mean the value of the network polynomial of $\mathcal{T}$ with input vector $\mathbf{x}$.

**Theorem 2.** If $\mathcal{T}$ is an induced tree from $\mathcal{S}$ over $X_{1:N}$, then $\mathcal{T}(\mathbf{x}) = \prod_{(v_i,v_j) \in \mathcal{T}_E} w_{ij} \prod_{n=1}^{N} \mathbb{I}_{x_n}$, where $w_{ij}$ is the edge weight of $(v_i, v_j)$ if $v_i$ is a sum node and $w_{ij} = 1$ if $v_i$ is a product node.

**Remark**. Although we focus our attention on Boolean random variables for the simplicity of discussion and illustration, Thm. 2 can be extended to the case where the univariate distributions at the leaf nodes are continuous or discrete distributions with countably infinitely many values, e.g., Gaussian distributions or Poisson distributions. We can simply replace the product of univariate distributions term, $\prod_{n=1}^{N} \mathbb{I}_{x_n}$, in Thm. 2 to be the general form $\prod_{n=1}^{N} p_n(X_n)$, where $p_n(X_n)$ is a univariate distribution over $X_n$. Also note that it is possible for two unique induced trees to share the same product of univariate distributions, but in this case their weight terms $\prod_{(v_i,v_i) \in \mathcal{T}_E} w_{ij}$ are guaranteed to be different. As we will see shortly, Thm. 2 implies that the joint distribution over $\{X_n\}_{n=1}^{N}$ represented by an SPN is essentially a mixture model with potentially exponentially many components in the mixture.

**Definition 3** (Network cardinality). The network cardinality $\tau_\mathcal{S}$ of an SPN $\mathcal{S}$ is the number of unique induced trees.

**Theorem 3.** $\tau_\mathcal{S} = f_\mathcal{S}(\mathbf{1}|\mathbf{1})$, where $f_\mathcal{S}(\mathbf{1}|\mathbf{1})$ is the value of the network polynomial of $\mathcal{S}$ with input vector $\mathbf{1}$ and all edge weights set to be 1.

**Theorem 4.** $\mathcal{S}(\mathbf{x}) = \sum_{t=1}^{\tau_\mathcal{S}} \mathcal{T}_t(\mathbf{x})$, where $\mathcal{T}_t$ is the $t$th unique induced tree of $\mathcal{S}$.

**Remark**. The above four theorems prove the fact that an SPN $\mathcal{S}$ is an ensemble or mixture of trees, where each tree computes an unnormalized distribution over $X_{1:N}$. The total number of unique trees in $\mathcal{S}$ is the network cardinality $\tau_\mathcal{S}$, which only depends on the structure of $\mathcal{S}$. Each component is a simple product of univariate distributions. We illustrate the theorems above with a simple example in Fig. 1.

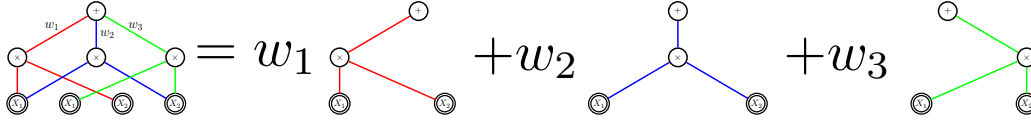

Figure 1: A complete and decomposable SPN is a mixture of induced trees. Double circles indicate univariate distributions over $X_1$ and $X_2$. Different colors are used to highlight unique induced trees; each induced tree is a product of univariate distributions over $X_1$ and $X_2$.

Zhao et al. [20] show that every complete and decomposable SPN is equivalent to a bipartite Bayesian network with a layer of hidden variables and a layer of observable random variables. The number of hidden variables in the bipartite Bayesian network is equal to the number of sum nodes in $\mathcal{S}$. A naive expansion of such Bayesian network to a mixture model will lead to a huge mixture model with $2^{O(M)}$ components, where $M$ is the number of sum nodes in $\mathcal{S}$. Here we complement their theory and show that each complete and decomposable SPN is essentially a mixture of trees and the effective number of unique induced trees is given by $\tau_\mathcal{S}$. Note that $\tau_\mathcal{S} = f_\mathcal{S}(\mathbf{1}|\mathbf{1})$ depends only on the network structure, and can often be much smaller than $2^{O(M)}$. Without loss of generality, assuming that in $\mathcal{S}$ layers of sum nodes are alternating with layers of product nodes, then $f_\mathcal{S}(\mathbf{1}|\mathbf{1}) = \Omega(2^h)$, where $h$ is the height of $\mathcal{S}$. However, the exponentially many trees are recursively merged and combined in $\mathcal{S}$ such that the overall network size is still tractable.

### 3.2 Maximum Likelihood Estimation as SP

Let's consider the likelihood function computed by an SPN $\mathcal{S}$ over $N$ binary random variables with model parameters $\mathbf{w}$ and input vector $\mathbf{x} \in \{0, 1\}^N$. Here the model parameters in $\mathcal{S}$ are edge weights from every sum node, and we collect them together into a long vector $\mathbf{w} \in \mathbb{R}_{++}^D$, where $D$ corresponds to the number of edges emanating from sum nodes in $\mathcal{S}$. By definition, the probability distribution induced by $\mathcal{S}$ can be computed by $\Pr_\mathcal{S}(\mathbf{x}|\mathbf{w}) \triangleq \frac{f_\mathcal{S}(\mathbf{x}|\mathbf{w})}{\sum_\mathbf{x} f_\mathcal{S}(\mathbf{x}|\mathbf{w})} = \frac{f_\mathcal{S}(\mathbf{x}|\mathbf{w})}{f_\mathcal{S}(\mathbf{1}|\mathbf{w})}$.

**Corollary 5.** Let $\mathcal{S}$ be an SPN with weights $\mathbf{w} \in \mathbb{R}_{++}^D$ over input vector $\mathbf{x} \in \{0, 1\}^N$, the network polynomial $f_\mathcal{S}(\mathbf{x}|\mathbf{w})$ is a posynomial: $f_\mathcal{S}(\mathbf{x}|\mathbf{w}) = \sum_{t=1}^{f_\mathcal{S}(\mathbf{1}|\mathbf{1})} \prod_{n=1}^N \mathbb{I}_{x_n}^{(t)} \prod_{d=1}^D w_d^{\mathbb{I}_{w_d \in \mathcal{T}_t}}$, where $\mathbb{I}_{w_d \in \mathcal{T}_t}$ is the indicator variable whether $w_d$ is in the $t$-th induced tree $\mathcal{T}_t$ or not. Each monomial corresponds exactly to a unique induced tree SPN from $\mathcal{S}$.

The above statement is a direct corollary of Thm. 2, Thm. 3 and Thm. 4. From the definition of network polynomial, we know that $f_\mathcal{S}$ is a multilinear function of the indicator variables. Corollary 5 works as a complement to characterize the functional form of a network polynomial in terms of $\mathbf{w}$. It follows that the likelihood function $\mathcal{L}_\mathcal{S}(\mathbf{w}) \triangleq \Pr_\mathcal{S}(\mathbf{x}|\mathbf{w})$ can be expressed as the ratio of two posynomial functions. We now show that the optimization problem based on MLE is an SP. Using the definition of $\Pr(\mathbf{x}|\mathbf{w})$ and Corollary 5, let $\tau = f_\mathcal{S}(\mathbf{1}|\mathbf{1})$, the MLE problem can be rewritten as

$$\text{maximize}_\mathbf{w} \quad \frac{f_\mathcal{S}(\mathbf{x}|\mathbf{w})}{f_\mathcal{S}(\mathbf{1}|\mathbf{w})} = \frac{\sum_{t=1}^\tau \prod_{n=1}^N \mathbb{I}_{x_n}^{(t)} \prod_{d=1}^D w_d^{\mathbb{I}_{w_d \in \mathcal{T}_t}}}{\sum_{t=1}^\tau \prod_{d=1}^D w_d^{\mathbb{I}_{w_d \in \mathcal{T}_t}}} \tag{1}$$

$$\text{subject to} \quad \mathbf{w} \in \mathbb{R}_{++}^D$$

**Proposition 6.** The MLE problem for SPNs is a signomial program.

Being nonconvex in general, SP is essentially hard to solve from a computational perspective [1, 3]. However, despite the hardness of SP in general, the objective function in the MLE formulation of SPNs has a special structure, i.e., it is the ratio of two posynomials, which makes the design of efficient optimization algorithms possible.

### 3.3 Difference of Convex Functions

Both PGD and EG are first-order methods and they can be viewed as approximating the SP after applying a logarithmic transformation to the objective function only. Although (1) is a signomial program, its objective function is expressed as the ratio of two posynomials. Hence, we can still apply the *logarithmic transformation trick* used in geometric programming to its objective function and to the variables to be optimized. More concretely, let $w_d = \exp(y_d), \forall d$ and take the $\log$ of the objective function; it becomes equivalent to maximize the following new objective without any constraint on $\mathbf{y}$:

$$\text{maximize} \quad \log \left( \sum_{t=1}^{\tau(\mathbf{x})} \exp \left( \sum_{d=1}^{D} y_d \mathbb{I}_{y_d \in \mathcal{T}_t} \right) \right) - \log \left( \sum_{t=1}^{\tau} \exp \left( \sum_{d=1}^{D} y_d \mathbb{I}_{y_d \in \mathcal{T}_t} \right) \right) \quad (2)$$

Note that in the first term of Eq. 2 the upper index $\tau(\mathbf{x}) \leq \tau \triangleq f_{\mathcal{S}}(\mathbf{1}|\mathbf{1})$ depends on the current input $\mathbf{x}$. By transforming into the log-space, we naturally guarantee the positivity of the solution at each iteration, hence transforming a constrained optimization problem into an unconstrained optimization problem without any sacrifice. Both terms in Eq. 2 are convex functions in $\mathbf{y}$ after the transformation. Hence, the transformed objective function is now expressed as the difference of two convex functions, which is called a DC function [9]. This helps us to design two efficient algorithms to solve the problem based on the general idea of sequential convex approximations for nonlinear programming.

#### 3.3.1 Sequential Monomial Approximation

Let's consider the linearization of both terms in Eq. 2 in order to apply first-order methods in the transformed space. To compute the gradient with respect to different components of $\mathbf{y}$, we view each node of an SPN as an intermediate function of the network polynomial and apply the chain rule to back-propagate the gradient. The differentiation of $f_{\mathcal{S}}(\mathbf{x}|\mathbf{w})$ with respect to the root node of the network is set to be 1. The differentiation of the network polynomial with respect to a partial function at each node can then be computed in two passes of the network: the bottom-up pass evaluates the values of all partial functions given the current input $\mathbf{x}$ and the top-down pass differentiates the network polynomial with respect to each partial function. Following the evaluation-differentiation passes, the gradient of the objective function in (2) can be computed in $O(|\mathcal{S}|)$. Furthermore, although the computation is conducted in $\mathbf{y}$, the results are fully expressed in terms of $\mathbf{w}$, which suggests that in practice we do not need to explicitly construct $\mathbf{y}$ from $\mathbf{w}$.

Let $f(\mathbf{y}) = \log f_{\mathcal{S}}(\mathbf{x}|\exp(\mathbf{y})) - \log f_{\mathcal{S}}(\mathbf{1}|\exp(\mathbf{y}))$. It follows that approximating $f(\mathbf{y})$ with the best linear function is equivalent to using the best monomial approximation of the signomial program (1). This leads to a sequential monomial approximations of the original SP formulation: at each iteration $\mathbf{y}^{(k)}$, we linearize both terms in Eq. 2 and form the optimal monomial function in terms of $\mathbf{w}^{(k)}$. The additive update of $\mathbf{y}^{(k)}$ leads to a multiplicative update of $\mathbf{w}^{(k)}$ since $\mathbf{w}^{(k)} = \exp(\mathbf{y}^{(k)})$, and we use a backtracking line search to determine the step size of the update in each iteration.

#### 3.3.2 Concave-convex Procedure

Sequential monomial approximation fails to use the structure of the problem when learning SPNs. Here we propose another approach based on the concave-convex procedure (CCCP) [18] to use the fact that the objective function is expressed as the difference of two convex functions. At a high level CCCP solves a sequence of concave surrogate optimizations until convergence. In many cases, the maximum of a concave surrogate function can only be solved using other convex solvers and as a result the efficiency of the CCCP highly depends on the choice of the convex solvers. However, we show that by a suitable transformation of the network we can compute the maximum of the concave surrogate in closed form in time that is linear in the network size, which leads to a very efficient

algorithm for learning the parameters of SPNs. We also prove the convergence properties of our algorithm.

Consider the objective function to be maximized in DCP: $f(\mathbf{y}) = \log f_{\mathcal{S}}(\mathbf{x}|\exp(\mathbf{y})) - \log f_{\mathcal{S}}(\mathbf{1}|\exp(\mathbf{y})) \triangleq f_1(\mathbf{y}) + f_2(\mathbf{y})$ where $f_1(\mathbf{y}) \triangleq \log f_{\mathcal{S}}(\mathbf{x}|\exp(\mathbf{y}))$ is a convex function and $f_2(\mathbf{y}) \triangleq -\log f_{\mathcal{S}}(\mathbf{1}|\exp(\mathbf{y}))$ is a concave function. We can linearize only the convex part $f_1(\mathbf{y})$ to obtain a surrogate function

$$\hat{f}(\mathbf{y}, \mathbf{z}) = f_1(\mathbf{z}) + \nabla_{\mathbf{z}} f_1(\mathbf{z})^T (\mathbf{y} - \mathbf{z}) + f_2(\mathbf{y}) \tag{3}$$

for $\forall \mathbf{y}, \mathbf{z} \in \mathbb{R}^D$. Now $\hat{f}(\mathbf{y}, \mathbf{z})$ is a concave function in $\mathbf{y}$. Due to the convexity of $f_1(\mathbf{y})$ we have $f_1(\mathbf{y}) \geq f_1(\mathbf{z}) + \nabla_{\mathbf{z}} f_1(\mathbf{z})^T(\mathbf{y} - \mathbf{z}), \forall \mathbf{y}, \mathbf{z}$ and as a result the following two properties always hold for $\forall \mathbf{y}, \mathbf{z}$: $\hat{f}(\mathbf{y}, \mathbf{z}) \leq f(\mathbf{y})$ and $\hat{f}(\mathbf{y}, \mathbf{y}) = f(\mathbf{y})$. CCCP updates $\mathbf{y}$ at each iteration $k$ by solving $\mathbf{y}^{(k)} \in \arg\max_{\mathbf{y}} \hat{f}(\mathbf{y}, \mathbf{y}^{(k-1)})$ unless we already have $\mathbf{y}^{(k-1)} \in \arg\max_{\mathbf{y}} \hat{f}(\mathbf{y}, \mathbf{y}^{(k-1)})$, in which case a generalized fixed point $\mathbf{y}^{(k-1)}$ has been found and the algorithm stops.

It is easy to show that at each iteration of CCCP we always have $f(\mathbf{y}^{(k)}) \geq f(\mathbf{y}^{(k-1)})$. Note also that $f(\mathbf{y})$ is computing the log-likelihood of input $\mathbf{x}$ and therefore it is bounded above by 0. By the monotone convergence theorem, $\lim_{k \to \infty} f(\mathbf{y}^{(k)})$ exists and the sequence $\{f(\mathbf{y}^{(k)})\}$ converges.

We now discuss how to compute a closed form solution for the maximization of the concave surrogate $\hat{f}(\mathbf{y}, \mathbf{y}^{(k-1)})$. Since $\hat{f}(\mathbf{y}, \mathbf{y}^{(k-1)})$ is differentiable and concave for any fixed $\mathbf{y}^{(k-1)}$, a sufficient and necessary condition to find its maximum is

$$\nabla_{\mathbf{y}} \hat{f}(\mathbf{y}, \mathbf{y}^{(k-1)}) = \nabla_{\mathbf{y}^{(k-1)}} f_1(\mathbf{y}^{(k-1)}) + \nabla_{\mathbf{y}} f_2(\mathbf{y}) = 0 \tag{4}$$

In the above equation, if we consider only the partial derivative with respect to $y_{ij}(w_{ij})$, we obtain

$$\frac{w_{ij}^{(k-1)} f_{v_j}(\mathbf{x}|\mathbf{w}^{(k-1)})}{f_{\mathcal{S}}(\mathbf{x}|\mathbf{w}^{(k-1)})} \frac{\partial f_{\mathcal{S}}(\mathbf{x}|\mathbf{w}^{(k-1)})}{\partial f_{v_i}(\mathbf{x}|\mathbf{w}^{(k-1)})} = \frac{w_{ij} f_{v_j}(\mathbf{1}|\mathbf{w})}{f_{\mathcal{S}}(\mathbf{1}|\mathbf{w})} \frac{\partial f_{\mathcal{S}}(\mathbf{1}|\mathbf{w})}{\partial f_{v_i}(\mathbf{1}|\mathbf{w})} \tag{5}$$

Eq. 5 leads to a system of $D$ nonlinear equations, which is hard to solve in closed form. However, if we do a change of variable by considering locally normalized weights $w'_{ij}$ (i.e., $w'_{ij} \geq 0$ and $\sum_j w'_{ij} = 1 \ \forall i$), then a solution can be easily computed. As described in [13, 20], any SPN can be transformed into an equivalent *normal* SPN with locally normalized weights in a bottom up pass as follows:

$$w'_{ij} = \frac{w_{ij} f_{v_j}(\mathbf{1}|\mathbf{w})}{\sum_j w_{ij} f_{v_j}(\mathbf{1}|\mathbf{w})} \tag{6}$$

We can then replace $w_{ij} f_{v_j}(\mathbf{1}|\mathbf{w})$ in the above equation by the expression it is equal to in Eq. 5 to obtain a closed form solution:

$$w'_{ij} \propto w_{ij}^{(k-1)} \frac{f_{v_j}(\mathbf{x}|\mathbf{w}^{(k-1)})}{f_{\mathcal{S}}(\mathbf{x}|\mathbf{w}^{(k-1)})} \frac{\partial f_{\mathcal{S}}(\mathbf{x}|\mathbf{w}^{(k-1)})}{\partial f_{v_i}(\mathbf{x}|\mathbf{w}^{(k-1)})} \tag{7}$$

Note that in the above derivation both $f_{v_i}(\mathbf{1}|\mathbf{w})/f_{\mathcal{S}}(\mathbf{1}|\mathbf{w})$ and $\partial f_{\mathcal{S}}(\mathbf{1}|\mathbf{w})/\partial f_{v_i}(\mathbf{1}|\mathbf{w})$ can be treated as constants and hence absorbed since $w'_{ij}, \forall j$ are constrained to be locally normalized. In order to obtain a solution to Eq. 5, for each edge weight $w_{ij}$, the sufficient statistics include only three terms, i.e, the evaluation value at $v_j$, the differentiation value at $v_i$ and the previous edge weight $w_{ij}^{(k-1)}$, all of which can be obtained in two passes of the network for each input $\mathbf{x}$. Thus the computational complexity to obtain a maximum of the concave surrogate is $O(|\mathcal{S}|)$. Interestingly, Eq. 7 leads to the same update formula as in the EM algorithm [12] despite the fact that CCCP and EM start from different perspectives. We show that all the limit points of the sequence $\{\mathbf{w}^{(k)}\}_{k=1}^{\infty}$ are guaranteed to be stationary points of DCP in (2).

**Theorem 7.** Let $\{\mathbf{w}^{(k)}\}_{k=1}^{\infty}$ be any sequence generated using Eq. 7 from any positive initial point, then all the limiting points of $\{\mathbf{w}^{(k)}\}_{k=1}^{\infty}$ are stationary points of the DCP in (2). In addition, $\lim_{k \to \infty} f(\mathbf{y}^{(k)}) = f(\mathbf{y}^*)$, where $\mathbf{y}^*$ is some stationary point of (2).

We summarize all four algorithms and highlight their connections and differences in Table 1. Although we mainly discuss the batch version of those algorithms, all of the four algorithms can be easily adapted to work in stochastic and/or parallel settings.

Table 1: Summary of PGD, EG, SMA and CCCP. Var. means the optimization variables.

| Algo | Var. | Update Type | Update Formula |
|---|---|---|---|
| PGD | $\mathbf{w}$ | Additive | $w_d^{(k+1)} \leftarrow P_{\mathbb{R}_{++}^\epsilon} \left\{ w_d^{(k)} + \gamma(\nabla_{w_d} f_1(\mathbf{w}^{(k)}) - \nabla_{w_d} f_2(\mathbf{w}^{(k)})) \right\}$ |
| EG | $\mathbf{w}$ | Multiplicative | $w_d^{(k+1)} \leftarrow w_d^{(k)} \exp\{\gamma(\nabla_{w_d} f_1(\mathbf{w}^{(k)}) - \nabla_{w_d} f_2(\mathbf{w}^{(k)}))\}$ |
| SMA | $\log \mathbf{w}$ | Multiplicative | $w_d^{(k+1)} \leftarrow w_d^{(k)} \exp\{\gamma w_d^{(k)} \times (\nabla_{w_d} f_1(\mathbf{w}^{(k)}) - \nabla_{w_d} f_2(\mathbf{w}^{(k)}))\}$ |
| CCCP | $\log \mathbf{w}$ | Multiplicative | $w_{ij}^{(k+1)} \propto w_{ij}^{(k)} \times \nabla_{v_i} f_{\mathcal{S}}(\mathbf{w}^{(k)}) \times f_{v_j}(\mathbf{w}^{(k)})$ |

## 4  Experiments

### 4.1  Experimental Setting

We conduct experiments on 20 benchmark data sets from various domains to compare and evaluate the convergence performance of the four algorithms: PGD, EG, SMA and CCCP (EM). These 20 data sets are widely used in [7, 15] to assess different SPNs for the task of density estimation. All the features in the 20 data sets are binary features. All the SPNs that are used for comparisons of PGD, EG, SMA and CCCP are trained using LearnSPN [7]. We discard the weights returned by LearnSPN and use random weights as initial model parameters. The random weights are determined by the same random seed in all four algorithms. Detailed information about these 20 datasets and the SPNs used in the experiments are provided in the supplementary material.

### 4.2  Parameter Learning

We implement all four algorithms in C++. For each algorithm, we set the maximum number of iterations to 50. If the absolute difference in the training log-likelihood at two consecutive steps is less than $0.001$, the algorithms are stopped. For PGD, EG and SMA, we combine each of them with backtracking line search and use a weight shrinking coefficient set at $0.8$. The learning rates are initialized to $1.0$ for all three methods. For PGD, we set the projection margin $\epsilon$ to $0.01$. There is no learning rate and no backtracking line search in CCCP. We set the smoothing parameter to $0.001$ in CCCP to avoid numerical issues.

We show in Fig. 2 the average log-likelihood scores on 20 training data sets to evaluate the convergence speed and stability of PGD, EG, SMA and CCCP. Clearly, CCCP wins by a large margin over PGD, EG and SMA, both in convergence speed and solution quality. Furthermore, among the four algorithms, CCCP is the most stable one due to its guarantee that the log-likelihood (on training data) will not decrease after each iteration. As shown in Fig. 2, the training curves of CCCP are more smooth than the other three methods in almost all the cases. These 20 experiments also clearly show that CCCP often converges in a few iterations. On the other hand, PGD, EG and SMA are on par with each other since they are all first-order methods. SMA is more stable than PGD and EG and often achieves better solutions than PGD and EG. On large data sets, SMA also converges faster than PGD and EG. Surprisingly, EG performs worse than PGD in some cases and is quite unstable despite the fact that it admits multiplicative updates. The "hook shape" curves of PGD in some data sets, e.g. Kosarak and KDD, are due to the projection operations.

Table 2: Average log-likelihoods on test data. Highest log-likelihoods are highlighted in bold. $\uparrow$ shows statistically better log-likelihoods than CCCP and $\downarrow$ shows statistically worse log-likelihoods than CCCP. The significance is measured based on the Wilcoxon signed-rank test.

| Data set | CCCP | LearnSPN | ID-SPN | Data set | CCCP | LearnSPN | ID-SPN |
|---|---|---|---|---|---|---|---|
| NLTCS | **-6.029** | ↓-6.099 | ↓-6.050 | DNA | -84.921 | ↓-85.237 | ↑**-84.693** |
| MSNBC | **-6.045** | ↓-6.113 | -6.048 | Kosarak | -10.880 | ↓-11.057 | **-10.605** |
| KDD 2k | **-2.134** | ↓-2.233 | ↓-2.153 | MSWeb | -9.970 | ↓-10.269 | **-9.800** |
| Plants | -12.872 | ↓-12.955 | ↑**-12.554** | Book | -35.009 | ↓-36.247 | ↑**-34.436** |
| Audio | -40.020 | ↓-40.510 | **-39.824** | EachMovie | -52.557 | ↓-52.816 | ↑**-51.550** |
| Jester | **-52.880** | ↓-53.454 | ↓-52.912 | WebKB | -157.492 | ↓-158.542 | ↑**-153.293** |
| Netflix | -56.782 | ↓-57.385 | ↑**-56.554** | Reuters-52 | -84.628 | ↓-85.979 | ↑**-84.389** |
| Accidents | -27.700 | ↓-29.907 | ↑**-27.232** | 20 Newsgrp | -153.205 | ↓-156.605 | ↑**-151.666** |
| Retail | **-10.919** | ↓-11.138 | -10.945 | BBC | **-248.602** | ↓-249.794 | ↓-252.602 |
| Pumsb-star | -24.229 | ↓-24.577 | ↑**-22.552** | Ad | **-27.202** | ↓-27.409 | ↓-40.012 |

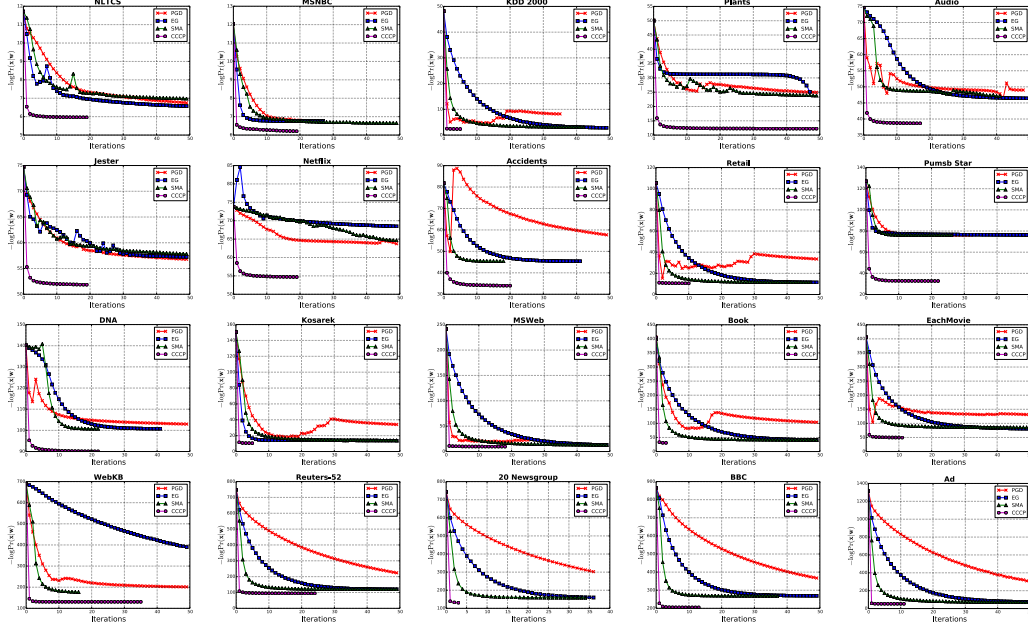

Figure 2: Negative log-likelihood values versus number of iterations for PGD, EG, SMA and CCCP.

The computational complexity per update is $O(|\mathcal{S}|)$ in all four algorithms. CCCP often takes less time than the other three algorithms because it takes fewer iterations to converge. We list detailed running time statistics for all four algorithms on the 20 data sets in the supplementary material.

## 4.3 Fine Tuning

We combine CCCP as a "fine tuning" procedure with the structure learning algorithm LearnSPN and compare it to the state-of-the-art structure learning algorithm ID-SPN [15]. More concretely, we keep the model parameters learned from LearnSPN and use them to initialize CCCP. We then update the model parameters globally using CCCP as a fine tuning technique. This normally helps to obtain a better generative model since the original parameters are learned greedily and locally during the structure learning algorithm. We use the validation set log-likelihood score to avoid overfitting. The algorithm returns the set of parameters that achieve the best validation set log-likelihood score as output. Experimental results are reported in Table. 2. As shown in Table 2, the use of CCCP after LearnSPN always helps to improve the model performance. By optimizing model parameters on these 20 data sets, we boost LearnSPN to achieve better results than state-of-the-art ID-SPN on 7 data sets, where the original LearnSPN only outperforms ID-SPN on 1 data set. Note that the sizes of the SPNs returned by LearnSPN are much smaller than those produced by ID-SPN. Hence, it is remarkable that by fine tuning the parameters with CCCP, we can achieve better performance despite the fact that the models are smaller. For a fair comparison, we also list the size of the SPNs returned by ID-SPN in the supplementary material.

## 5 Conclusion

We show that the network polynomial of an SPN is a posynomial function of the model parameters, and that parameter learning yields a signomial program. We propose two convex relaxations to solve the SP. We analyze the convergence properties of CCCP for learning SPNs. Extensive experiments are conducted to evaluate the proposed approaches and current methods. We also recommend combining CCCP with structure learning algorithms to boost the modeling accuracy.

**Acknowledgments**

HZ and GG gratefully acknowledge support from ONR contract N000141512365. HZ also thanks Ryan Tibshirani for the helpful discussion about CCCP.

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
