[Supplementary Material]

# A  Proof of SPNs as Mixture of Trees

**Theorem 1.** *If $\mathcal{T}$ is an induced SPN from a complete and decomposable SPN $\mathcal{S}$, then $\mathcal{T}$ is a tree that is complete and decomposable.*

*Proof.* Argue by contradiction that $\mathcal{T}$ is not a tree, then there must exist a node $v \in \mathcal{T}$ such that $v$ has more than one parent in $\mathcal{T}$. This means that there exist at least two paths $R, p_1, \ldots, v$ and $R, q_1, \ldots, v$ that connect the root of $\mathcal{S}(\mathcal{T})$, which we denote by $R$, and $v$. Let $t$ be the last node in $R, p_1, \ldots, v$ and $R, q_1, \ldots, v$ such that $R, \ldots, t$ are common prefix of these two paths. By construction we know that such $t$ must exist since these two paths start from the same root node $R$ ($R$ will be one candidate of such $t$). Also, we claim that $t \neq v$ otherwise these two paths overlap with each other, which contradicts the assumption that $v$ has multiple parents. This shows that these two paths can be represented as $R, \ldots, t, p, \ldots, v$ and $R, \ldots, t, q, \ldots, v$ where $R, \ldots, t$ are the common prefix shared by these two paths and $p \neq q$ since $t$ is the last common node. From the construction process defined in Def. 2, we know that both $p$ and $q$ are children of $t$ in $\mathcal{S}$. Recall that for each sum node in $\mathcal{S}$, Def. 2 takes at most one child, hence we claim that $t$ must be a product node, since both $p$ and $q$ are children of $t$. Then the paths that $t \to p \rightsquigarrow v$ and $t \to q \rightsquigarrow v$ indicate that $\mathrm{scope}(v) \subseteq \mathrm{scope}(p) \subseteq \mathrm{scope}(t)$ and $\mathrm{scope}(v) \subseteq \mathrm{scope}(q) \subseteq \mathrm{scope}(t)$, leading to $\varnothing \neq \mathrm{scope}(v) \subseteq \mathrm{scope}(p) \cap \mathrm{scope}(q)$, which is a contradiction of the decomposability of the product node $t$. Hence as long as $\mathcal{S}$ is complete and decomposable, $\mathcal{T}$ must be a tree.

The completeness of $\mathcal{T}$ is trivially satisfied because each sum node has only one child in $\mathcal{T}$. It is also straightforward to verify that $\mathcal{T}$ satisfies the decomposability as $\mathcal{T}$ is an induced subgraph of $\mathcal{S}$, which is decomposable. ∎

**Theorem 2.** *If $\mathcal{T}$ is an induced tree from $\mathcal{S}$ over $X_{1:N}$, then $\mathcal{T}(\mathbf{x}) = \prod_{(v_i, v_j) \in \mathcal{T}_E} w_{ij} \prod_{n=1}^{N} \mathbb{I}_{x_n}$, where $w_{ij}$ is the edge weight of $(v_i, v_j)$ if $v_i$ is a sum node and $w_{ij} = 1$ if $v_i$ is a product node.*

*Proof.* First, the scope of $\mathcal{T}$ is the same as the scope of $\mathcal{S}$ because the root of $\mathcal{S}$ is also the root of $\mathcal{T}$. This shows that for each $X_i$ there is at least one indicator $\mathbb{I}_{x_i}$ in the leaves otherwise the scope of the root node of $\mathcal{T}$ will be a strict subset of the scope of the root node of $\mathcal{S}$. Furthermore, for each variable $X_i$ there is at most one indicator $\mathbb{I}_{x_i}$ in the leaves. This is observed by the fact that there is at most one child collected from a sum node into $\mathcal{T}$ and if $\mathbb{I}_{x_i}$ and $\mathbb{I}_{\bar{x}_i}$ appear simultaneously in the leaves, then their least common ancestor must be a product node. Note that the least common ancestor of $\mathbb{I}_{x_i}$ and $\mathbb{I}_{\bar{x}_i}$ is guaranteed to exist because of the tree structure of $\mathcal{T}$. However, this leads to a contradiction of the fact that $\mathcal{S}$ is decomposable. As a result, there is exactly one indicator $\mathbb{I}_{x_i}$ for each variable $X_i$ in $\mathcal{T}$. Hence the multiplicative constant of the monomial admits the form $\prod_{i=1}^{n} \mathbb{I}_{x_i}$, which is a product of univariate distributions. More specifically, it is a product of indicator variables in the case of Boolean input variables.

We have already shown that $\mathcal{T}$ is a tree and only product nodes in $\mathcal{T}$ can have multiple children. It follows that the functional form of $f_{\mathcal{T}}(\mathbf{x})$ must be a monomial, and only edge weights that are in $\mathcal{T}$ contribute to the monomial. Combing all the above, we know that $f_{\mathcal{T}}(\mathbf{x}) = \prod_{(v_i, v_i) \in \mathcal{T}_E} w_{ij} \prod_{n=1}^{N} \mathbb{I}_{x_n}$. ∎

**Theorem 3.** $\tau_{\mathcal{S}} = f_{\mathcal{S}}(\mathbf{1}|\mathbf{1})$, *where $f_{\mathcal{S}}(\mathbf{1}|\mathbf{1})$ is the value of the network polynomial of $\mathcal{S}$ with input vector $\mathbf{1}$ and all edge weights set to be 1.*

**Theorem 4.** $\mathcal{S}(\mathbf{x}) = \sum_{t=1}^{\tau_{\mathcal{S}}} \mathcal{T}_t(\mathbf{x})$, *where $\mathcal{T}_t$ is the $t$th unique induced tree of $\mathcal{S}$.*

*Proof.* We prove by induction on the height of $\mathcal{S}$. If the height of $\mathcal{S}$ is 2, then depending on the type of the root node, we have two cases:

1. If the root is a sum node with $K$ children, then there are $C_K^1 = K$ different subgraphs that satisfy Def. 2, which is exactly the value of the network by setting all the indicators and edge weights from the root to be 1.
2. If the root is a product node then there is only 1 subgraph which is the graph itself. Again, this equals to the value of $\mathcal{S}$ by setting all indicators to be 1.

Assume the theorem is true for SPNs with height $\leq h$. Consider an SPN $\mathcal{S}$ with height $h + 1$. Again, depending on the type of the root node, we need to discuss two cases:

1.  If the root is a sum node with $K$ children, where the $k$th sub-SPN has $f_{\mathcal{S}_k}(\mathbf{1}|\mathbf{1})$ unique induced trees, then by Def. 2 the total number of unique induced trees of $\mathcal{S}$ is $\sum_{k=1}^{K} f_{\mathcal{S}_k}(\mathbf{1}|\mathbf{1}) = \sum_{k=1}^{K} 1 \cdot f_{\mathcal{S}_k}(\mathbf{1}|\mathbf{1}) = f_{\mathcal{S}}(\mathbf{1}|\mathbf{1})$.
2.  If the root is a product node with $K$ children, then the total number of unique induced trees of $\mathcal{S}$ can then be computed by $\prod_{k=1}^{K} f_{\mathcal{S}_k}(\mathbf{1}|\mathbf{1}) = f_{\mathcal{S}}(\mathbf{1}|\mathbf{1})$.

The second part of the theorem follows by using distributive law between multiplication and addition to combine unique trees that share the same prefix in bottom-up order. ∎

## B  MLE as Signomial Programming

**Proposition 6.** The MLE problem for SPNs is a signomial program.

*Proof.* Using the definition of $\Pr(\mathbf{x}|\mathbf{w})$ and Corollary 5, let $\tau = f_{\mathcal{S}}(\mathbf{1}|\mathbf{1})$, the MLE problem can be rewritten as

$$
\begin{aligned}
\text{maximize}_{\mathbf{w}} \quad & \frac{f_{\mathcal{S}}(\mathbf{x}|\mathbf{w})}{f_{\mathcal{S}}(\mathbf{1}|\mathbf{w})} = \frac{\sum_{t=1}^{\tau} \prod_{n=1}^{N} \mathbb{I}_{x_n}^{(t)} \prod_{d=1}^{D} w_d^{\mathbb{I}_{w_d \in \mathcal{T}_t}}}{\sum_{t=1}^{\tau} \prod_{d=1}^{D} w_d^{\mathbb{I}_{w_d \in \mathcal{T}_t}}} \\
\text{subject to} \quad & \mathbf{w} \in \mathbb{R}_{++}^{D}
\end{aligned}
\tag{8}
$$

which we claim is equivalent to:

$$
\begin{aligned}
\text{minimize}_{\mathbf{w}, z} \quad & -z \\
\text{subject to} \quad & \sum_{t=1}^{\tau} z \prod_{d=1}^{D} w_d^{\mathbb{I}_{w_d \in \mathcal{T}_t}} - \sum_{l=1}^{\tau} \prod_{n=1}^{N} \mathbb{I}_{x_n}^{(t)} \prod_{d=1}^{D} w_d^{\mathbb{I}_{w_d \in \mathcal{T}_t}} \le 0 \\
& \mathbf{w} \in \mathbb{R}_{++}^{D}, z > 0
\end{aligned}
\tag{9}
$$

It is easy to check that both the objective function and constraint function in (9) are signomials. To see the equivalence of (8) and (9), let $p^*$ be the optimal value of (8) achieved at $\mathbf{w}^*$. Choose $z = p^*$ and $\mathbf{w} = \mathbf{w}^*$ in (9), then $-z$ is also the optimal solution of (9) otherwise we can find feasible $(z', \mathbf{w}')$ in (9) which has $-z' < -z \Leftrightarrow z' > z$. Combined with the constraint function in (9), we have $p^* = z < z' \le \frac{f_{\mathcal{S}}(\mathbf{x}|\mathbf{w}')}{f_{\mathcal{S}}(\mathbf{1}|\mathbf{w}')}$, which contradicts the optimality of $p^*$. In the other direction, let $z^*, \mathbf{w}^*$ be the solution that achieves optimal value of (9), then we claim that $z^*$ is also the optimal value of (8), otherwise there exists a feasible $\mathbf{w}$ in (8) such that $z \triangleq \frac{f_{\mathcal{S}}(\mathbf{x}|\mathbf{w})}{f_{\mathcal{S}}(\mathbf{1}|\mathbf{w})} > z^*$. Since $(\mathbf{w}, z)$ is also feasible in (9) with $-z < -z^*$, this contradicts the optimality of $z^*$. ∎

The transformation from (8) to (9) does not make the problem any easier to solve. Rather, it destroys the structure of (8), i.e., the objective function of (8) is the ratio of two posynomials. However, the equivalent transformation does reveal some insights about the intrinsic complexity of the optimization problem, which indicates that it is hard to solve (8) efficiently with the guarantee of achieving a globally optimal solution.

## C  Convergence of CCCP for SPNs

We discussed before that the sequence of function values $\{f(\mathbf{y}^{(k)})\}$ converges to a limiting point. However, this fact alone does not necessarily indicate that $\{f(\mathbf{y}^{(k)})\}$ converges to $f(\mathbf{y}^*)$ where $\mathbf{y}^*$ is a stationary point of $f(\cdot)$ nor does it imply that the sequence $\{\mathbf{y}^{(k)}\}$ converges as $k \to \infty$. *Zangwill's global convergence theory* [19] has been successfully applied to study the convergence properties of many iterative algorithms frequently used in machine learning, including EM [17], generalized alternating minimization [8] and also CCCP [11]. Here we also apply Zangwill's theory and combine the analysis from [11] to show the following theorem:

**Theorem 7.** Let $\{\mathbf{w}^{(k)}\}_{k=1}^{\infty}$ be any sequence generated using Eq. 7 from any positive initial point, then all the limiting points of $\{\mathbf{w}^{(k)}\}_{k=1}^{\infty}$ are stationary points of the DCP in (2). In addition, $\lim_{k \to \infty} f(\mathbf{y}^{(k)}) = f(\mathbf{y}^*)$, where $\mathbf{y}^*$ is some stationary point of (2).

*Proof.* We will use Zangwill's global convergence theory for iterative algorithms [19] to show the convergence in our case. Before showing the proof, we need to first introduce the notion of "point-to-set mapping", where the output of the mapping is defined to be a set. More formally, a point-to-set map $\Phi$ from a set $\mathcal{X}$ to $\mathcal{Y}$ is defined as $\Phi : \mathcal{X} \mapsto \mathcal{P}(\mathcal{Y})$, where $\mathcal{P}(\mathcal{Y})$ is the power set of $\mathcal{Y}$. Suppose $\mathcal{X}$ and $\mathcal{Y}$ are equipped with the norm $||\cdot||_{\mathcal{X}}$ and $||\cdot||_{\mathcal{Y}}$, respectively. A point-to-set map $\Phi$ is said to be *closed* at $x^* \in \mathcal{X}$ if $x_k \in \mathcal{X}$, $\{x_k\}_{k=1}^{\infty} \to x^*$ and $y_k \in \mathcal{Y}$, $\{y_k\}_{k=1}^{\infty} \to y^*$, $y_k \in \Phi(x_k)$ imply that $y^* \in \Phi(x^*)$. A point-to-set map $\Phi$ is said to be closed on $S \subseteq \mathcal{X}$ if $\Phi$ is closed at every point in $S$. The concept of *closedness* in the point-to-set map setting reduces to *continuity* if we restrict that the output of $\Phi$ to be a set of singleton for every possible input, i.e., when $\Phi$ is a point-to-point mapping.

**Theorem 8** (Global Convergence Theorem [19])**.** Let the sequence $\{x_k\}_{k=1}^{\infty}$ be generated by $x_{k+1} \in \Phi(x_k)$, where $\Phi(\cdot)$ is a point-to-set map from $\mathcal{X}$ to $\mathcal{X}$. Let a solution set $\Gamma \subseteq \mathcal{X}$ be given, and suppose that:

1. all points $x_k$ are contained in a compact set $S \subseteq \mathcal{X}$.
2. $\Phi$ is closed over the complement of $\Gamma$.
3. there is a continuous function $\alpha$ on $\mathcal{X}$ such that:
    (a) if $x \notin \Gamma$, $\alpha(x') > \alpha(x)$ for $\forall x' \in \Phi(x)$.
    (b) if $x \in \Gamma$, $\alpha(x') \geq \alpha(x)$ for $\forall x' \in \Phi(x)$.

Then all the limit points of $\{x_k\}_{k=1}^{\infty}$ are in the solution set $\Gamma$ and $\alpha(x_k)$ converges monotonically to $\alpha(x^*)$ for some $x^* \in \Gamma$.

Let $\mathbf{w} \in \mathbb{R}_+^D$. Let $\Phi(\mathbf{w}^{(k-1)}) = \exp(\arg\max_{\mathbf{y}} \hat{f}(\mathbf{y}, \mathbf{y}^{(k-1)}))$ and let $\alpha(\mathbf{w}) = f(\log \mathbf{w}) = f(\mathbf{y}) = \log f_{\mathcal{S}}(\mathbf{x}|\exp(\mathbf{y})) - \log f_{\mathcal{S}}(\mathbf{1}|\exp(\mathbf{y}))$. Here we use $\mathbf{w}$ and $\mathbf{y}$ interchangeably as $\mathbf{w} = \exp(\mathbf{y})$ or each component is a one-to-one mapping. Note that since the $\arg\max_{\mathbf{y}} \hat{f}(\mathbf{y}, \mathbf{y}^{(k-1)})$ given $\mathbf{y}^{(k-1)}$ is achievable, $\Phi(\cdot)$ is a well defined point-to-set map for $\mathbf{w} \in \mathbb{R}_+^D$.

Specifically, in our case given $\mathbf{w}^{(k-1)}$, at each iteration of Eq. 7 we have

$$w'_{ij} = \frac{w_{ij} f_{v_j}(\mathbf{1}|\mathbf{w})}{\sum_j w_{ij} f_{v_j}(\mathbf{1}|\mathbf{w})} \propto w_{ij}^{(k-1)} \frac{f_{v_j}(\mathbf{x}|\mathbf{w}^{(k-1)})}{f_{\mathcal{S}}(\mathbf{x}|\mathbf{w}^{(k-1)})} \frac{\partial f_{\mathcal{S}}(\mathbf{x}|\mathbf{w}^{(k-1)})}{\partial f_{v_i}(\mathbf{x}|\mathbf{w}^{(k-1)})}$$

i.e., the point-to-set mapping is given by

$$\Phi_{ij}(\mathbf{w}^{(k-1)}) = \frac{w_{ij}^{(k-1)} f_{v_j}(\mathbf{x}|\mathbf{w}^{(k-1)}) \frac{\partial f_{\mathcal{S}}(\mathbf{x}|\mathbf{w}^{(k-1)})}{\partial f_{v_i}(\mathbf{x}|\mathbf{w}^{(k-1)})}}{\sum_{j'} w_{ij'}^{(k-1)} f_{v_{j'}}(\mathbf{x}|\mathbf{w}^{(k-1)}) \frac{\partial f_{\mathcal{S}}(\mathbf{x}|\mathbf{w}^{(k-1)})}{\partial f_{v_i}(\mathbf{x}|\mathbf{w}^{(k-1)})}}$$

Let $S = [0,1]^D$, the $D$ dimensional hyper cube. Then the above update formula indicates that $\Phi(\mathbf{w}^{(k-1)}) \in S$. Furthermore, if we assume $\mathbf{w}^{(1)} \in S$, which can be obtained by local normalization before any update, we can guarantee that $\{\mathbf{w}_k\}_{k=1}^{\infty} \subseteq S$, which is a compact set in $\mathbb{R}_+^D$.

The solution to $\max_{\mathbf{y}} \hat{f}(\mathbf{y}, \mathbf{y}^{(k-1)})$ is not unique. In fact, there are infinitely many solutions to this nonlinear equations. However, as we define above, $\Phi(\mathbf{w}^{(k-1)})$ returns *one* solution to this convex program in the $D$ dimensional hyper cube. Hence in our case $\Phi(\cdot)$ reduces to a point-to-point map, where the definition of *closedness* of a point-to-set map reduces to the notion of *continuity* of a point-to-point map. Define $\Gamma = \{\mathbf{w}^* \mid \mathbf{w}^*$ is a stationary point of $\alpha(\cdot)\}$. Hence we only need to verify the continuity of $\Phi(\mathbf{w})$ when $\mathbf{w} \in S$. To show this, we first characterize the functional form of $\frac{\partial f_{\mathcal{S}}(\mathbf{x}|\mathbf{w})}{\partial f_{v_i}(\mathbf{x}|\mathbf{w})}$ as it is used inside $\Phi(\cdot)$. We claim that for each node $v_i$, $\frac{\partial f_{\mathcal{S}}(\mathbf{x}|\mathbf{w})}{\partial f_{v_i}(\mathbf{x}|\mathbf{w})}$ is again, a posynomial function of $\mathbf{w}$. A graphical illustration is given in Fig. 3 to explain the process. This can also be derived from the sum rules and product rules used during top-down differentiation. More specifically, if $v_i$ is a product node, let $v_j, j = 1, \ldots, J$ be its parents in the network, which are assumed to be sum nodes, the differentiation of $f_{\mathcal{S}}$ with respect to $f_{v_i}$ is given by $\frac{\partial f_{\mathcal{S}}(\mathbf{x}|\mathbf{w})}{\partial f_{v_i}(\mathbf{x}|\mathbf{w})} = \sum_{j=1}^J \frac{\partial f_{\mathcal{S}}(\mathbf{x}|\mathbf{w})}{\partial f_{v_j}(\mathbf{x}|\mathbf{w})} \frac{\partial f_{v_j}(\mathbf{x}|\mathbf{w})}{\partial f_{v_i}(\mathbf{x}|\mathbf{w})}$. We reach

$$\frac{\partial f_{\mathcal{S}}(\mathbf{x}|\mathbf{w})}{\partial f_{v_i}(\mathbf{x}|\mathbf{w})} = \sum_{j=1}^J w_{ij} \frac{\partial f_{\mathcal{S}}(\mathbf{x}|\mathbf{w})}{\partial f_{v_j}(\mathbf{x}|\mathbf{w})} \tag{10}$$

Figure 3: Graphical illustration of $\frac{\partial f_S(\mathbf{x}|\mathbf{w})}{\partial f_{v_i}(\mathbf{x}|\mathbf{w})}$. The partial derivative of $f_S$ with respect to $f_{v_i}$ (in red) is a posynomial that is a product of edge weights lying on the path from root to $v_i$ and network polynomials from nodes that are children of product nodes on the path (highlighted in blue).

Similarly, if $v_i$ is a sum node and its parents $v_j, j = 1, \ldots, J$ are assumed to be product nodes, we have

$$\frac{\partial f_S(\mathbf{x}|\mathbf{w})}{\partial f_{v_i}(\mathbf{x}|\mathbf{w})} = \sum_{j=1}^{J} \frac{\partial f_S(\mathbf{x}|\mathbf{w})}{\partial f_{v_j}(\mathbf{x}|\mathbf{w})} \frac{f_{v_j}(\mathbf{x}|\mathbf{w})}{f_{v_i}(\mathbf{x}|\mathbf{w})} \tag{11}$$

Since $v_j$ is a product node and $v_j$ is a parent of $v_i$, so the last term in Eq. 11 can be equivalently expressed as

$$\frac{f_{v_j}(\mathbf{x}|\mathbf{w})}{f_{v_i}(\mathbf{x}|\mathbf{w})} = \prod_{h \neq i} f_{v_h}(\mathbf{x}|\mathbf{w})$$

where the index is range from all the children of $v_j$ except $v_i$. Combining the fact that the partial differentiation of $f_S$ with respect to the root node is 1 and that each $f_v$ is a posynomial function, it follows by induction in top-down order that $\frac{\partial f_S(\mathbf{x}|\mathbf{w})}{\partial f_{v_i}(\mathbf{x}|\mathbf{w})}$ is also a posynomial function of $\mathbf{w}$.

We have shown that both the numerator and the denominator of $\Phi(\cdot)$ are posynomial functions of $\mathbf{w}$. Because posynomial functions are continuous functions, in order to show that $\Phi(\cdot)$ is also continuous on $S\backslash\Gamma$, we need to guarantee that the denominator is not a degenerate posynomial function, i.e., the denominator of $\Phi(\mathbf{w}) \neq 0$ for all possible input vector $\mathbf{x}$. Recall that $\Gamma = \{\mathbf{w}^* \mid \mathbf{w}^* \text{ is a stationary point of } \alpha(\cdot)\}$, hence $\forall \mathbf{w} \in S\backslash\Gamma$, $\mathbf{w} \notin$ bd $S$, where bd $S$ is the boundary of the $D$ dimensional hyper cube $S$. Hence we have $\forall \mathbf{w} \in S\backslash\Gamma \Rightarrow \mathbf{w} \in$ int $S \Rightarrow \mathbf{w} > 0$ for each component. This immediately leads to $f_v(\mathbf{x}|\mathbf{w}) > 0, \forall v$. As a result, $\Phi(\mathbf{w})$ is continuous on $S\backslash\Gamma$ since it is the ratio of two strictly positive posynomial functions.

We now verify the third property in Zangwill's global convergence theory. At each iteration of CCCP, we have the following two cases to consider:

1. If $\mathbf{w}^{(k-1)} \notin \Gamma$, i.e., $\mathbf{w}^{(k-1)}$ is not a stationary point of $\alpha(\mathbf{w})$, then $\mathbf{y}^{(k-1)} \notin \arg\max_{\mathbf{y}} \hat{f}(\mathbf{y}, \mathbf{y}^{(k-1)})$, so we have $\alpha(\mathbf{w}^{(k)}) = f(\mathbf{y}^{(k)}) \geq \hat{f}(\mathbf{y}^{(k)}, \mathbf{y}^{(k-1)}) > \hat{f}(\mathbf{y}^{(k-1)}, \mathbf{y}^{(k-1)}) = f(\mathbf{y}^{(k-1)}) = \alpha(\mathbf{w}^{(k-1)})$.
2. If $\mathbf{w}^{(k-1)} \in \Gamma$, i.e., $\mathbf{w}^{(k-1)}$ is a stationary point of $\alpha(\mathbf{w})$, then $\mathbf{y}^{(k-1)} \in \arg\max_{\mathbf{y}} \hat{f}(\mathbf{y}, \mathbf{y}^{(k-1)})$, so we have $\alpha(\mathbf{w}^{(k)}) = f(\mathbf{y}^{(k)}) \geq \hat{f}(\mathbf{y}^{(k)}, \mathbf{y}^{(k-1)}) = \hat{f}(\mathbf{y}^{(k-1)}, \mathbf{y}^{(k-1)}) = f(\mathbf{y}^{(k-1)}) = \alpha(\mathbf{w}^{(k-1)})$.

By Zangwill's global convergence theory, we now conclude that all the limit points of $\{\mathbf{w}_k\}_{k=1}^{\infty}$ are in $\Gamma$ and $\alpha(\mathbf{w}_k)$ converges monotonically to $\alpha(\mathbf{w}^*)$ for some stationary point $\mathbf{w}^* \in \Gamma$. ∎

**Remark 1.** Technically we need to choose $\mathbf{w}_0 \in$ int $S$ to ensure the continuity of $\Phi(\cdot)$. This initial condition combined with the fact that inside each iteration of CCCP the algorithm only applies

Figure 4: A counterexample of SPN over two binary random variables where the weights $w_1, w_2, w_3$ are symmetric and indistinguishable.

positive multiplicative update and renormalization, ensure that after any finite $k$ steps, $\mathbf{w}_k \in \text{int}S$. Theoretically, only in the limit it is possible that some components of $\mathbf{w}_\infty$ may become 0. However in practice, due to the numerical precision of float numbers on computers, it is possible that after some finite update steps some of the components in $\mathbf{w}_k$ become 0. So in practical implementation we recommend to use a small positive number $\epsilon$ to smooth out such 0 components in $\mathbf{w}_k$ during the iterations of CCCP. Such smoothing may hurt the monotonic property of CCCP, but this can only happens when $\mathbf{w}_k$ is close to $\mathbf{w}^*$ and we can use early stopping to obtain a solution in the interior of $S$.

**Remark 2.** Thm. 7 only implies that any limiting point of the sequence $\{\mathbf{w}_k\}_{k=1}^\infty(\{\mathbf{y}_k\}_{k=1}^\infty)$ must be a stationary point of the log-likelihood function and $\{f(\mathbf{y})_k\}_{k=1}^\infty$ must converge to some $f(\mathbf{y}^*)$ where $\mathbf{y}^*$ is a stationary point. Thm. 7 does not imply that the sequence $\{\mathbf{w}_k\}_{k=1}^\infty(\{\mathbf{y}_k\}_{k=1}^\infty)$ is guaranteed to converge. [11] studies the convergence property of general CCCP procedure. Under more strong conditions, i.e., the strict concavity of the surrogate function or that $\Phi()$ to be a contraction mapping, it is possible to show that the sequence $\{\mathbf{w}_k\}_{k=1}^\infty(\{\mathbf{y}_k\}_{k=1}^\infty)$ also converges. However, none of such conditions hold in our case. In fact, in general there are infinitely many fixed points of $\Phi(\cdot)$, i.e., the equation $\Phi(\mathbf{w}) = \mathbf{w}$ has infinitely many solutions in $S$. Also, for a fixed value $t$, if $\alpha(\mathbf{w}) = t$ has at least one solution, then there are infinitely many solutions. Such properties of SPNs make it generally very hard to guarantee the convergence of the sequence $\{\mathbf{w}_k\}_{k=1}^\infty(\{\mathbf{y}_k\}_{k=1}^\infty)$. We give a very simple example below to illustrate the hardness in SPNs in Fig. 4. Consider applying the CCCP procedure to learn the parameters on the SPN given in Fig. 4 with three instances $\{(0,1),(1,0),(1,1)\}$. Then if we choose the initial parameter $\mathbf{w}_0$ such that the weights over the indicator variables are set as shown in Fig. 4, then any assignment of $(w_1, w_2, w_3)$ in the probability simplex will be equally optimal in terms of likelihood on inputs. In this example, there are uncountably infinite equal solutions, which invalidates the finite solution set requirement given in [11] in order to show the convergence of $\{\mathbf{w}_k\}_{k=1}^\infty$. However, we emphasize that the convergence of the sequence $\{\mathbf{w}_k\}_{k=1}^\infty$ is not as important as the convergence of $\{\alpha(\mathbf{w})_k\}_{k=1}^\infty$ to desired locations on the log-likelihood surface as in practice any $\mathbf{w}^*$ with equally good log-likelihood may suffice for the inference/prediction task.

It is worth to point out that the above theorem *does not* imply the convergence of the sequence $\{\mathbf{w}^{(k)}\}_{k=1}^\infty$. Thm. 7 only indicates that all the limiting points of $\{\mathbf{w}^{(k)}\}_{k=1}^\infty$, i.e., the limits of subsequences of $\{\mathbf{w}^{(k)}\}_{k=1}^\infty$, are stationary points of the DCP in (2). We also present a negative example in Fig. 4 that invalidates the application of Zangwill's global convergence theory on the analysis in this case.

The convergence rate of general CCCP is still an open problem [11]. [16] studied the convergence rate of unconstrained bound optimization algorithms with differentiable objective functions, of which our problem is a special case. The conclusion is that depending on the curvature of $f_1$ and $f_2$ (which are functions of the training data), CCCP will exhibit either a quasi-Newton behavior with superlinear convergence or first-order convergence. We show in experiments that CCCP normally exhibits a fast,

superlinear convergence rate compared with PGD, EG and SMA. Both CCCP and EM are special cases of a more general framework known as Majorization-Maximization. We show that in the case of SPNs these two algorithms coincide with each other, i.e., they lead to the same update formulas despite the fact that they start from totally different perspectives.

# D Experiment Details

## D.1 Methods

We will briefly review the current approach for training SPNs using projected gradient descent (PGD). Another related approach is to use exponentiated gradient (EG) [10] to optimize (8). PGD optimizes the log-likelihood by projecting the intermediate solution back to the positive orthant after each gradient update. Since the constraint in (8) is an open set, we need to manually create a closed set on which the projection operation can be well defined. One feasible choice is to project on to $\mathbb{R}^D_\epsilon \triangleq \{\mathbf{w} \in \mathbb{R}^D_{++} \mid w_d \geq \epsilon, \forall d\}$ where $\epsilon > 0$ is assumed to be very small. To avoid the projection, one direct solution is to use the exponentiated gradient (EG) method[10], which was first applied in an online setting and latter successfully extended to batch settings when training with convex models. EG admits a multiplicative update at each iteration and hence avoids the need for projection in PGD. However, EG is mostly applied in convex setting and it is not clear whether the convergence guarantee still holds or not in nonconvex setting.

## D.2 Experimental Setup

The sizes of different SPNs produced by LearnSPN and ID-SPN are shown in Table 3.

Table 3: Sizes of SPNs produced by LearnSPN and ID-SPN.

| Data set | LearnSPN | ID-SPN |
|----------|----------|--------|
| NLTCS | 13,733 | 24,690 |
| MSNBC | 54,839 | 579,364 |
| KDD 2k | 48,279 | 1,286,657 |
| Plants | 132,959 | 2,063,708 |
| Audio | 739,525 | 2,643,948 |
| Jester | 314,013 | 4,225,471 |
| Netflix | 161,655 | 7,958,088 |
| Accidents | 204,501 | 2,273,186 |
| Retail | 56,931 | 60,961 |
| Pumsb-star | 140,339 | 1,751,092 |
| DNA | 108,021 | 3,228,616 |
| Kosarak | 203,321 | 1,272,981 |
| MSWeb | 68,853 | 1,886,777 |
| Book | 190,625 | 1,445,501 |
| EachMovie | 522,753 | 2,440,864 |
| WebKB | 1,439,751 | 2,605,141 |
| Reuters-52 | 2,210,325 | 4,563,861 |
| 20 Newsgrp | 14,561,965 | 3,485,029 |
| BBC | 1,879,921 | 2,426,602 |
| Ad | 4,133,421 | 2,087,253 |

We list here the detailed statistics of the 20 data sets used in the experiments in Table 4. Table 5 shows the detailed running time of PGD, EG, SMA and CCCP on 20 data sets, measured in seconds.

Table 4: Statistics of data sets and models. $N$ is the number of variables modeled by the network, $|\mathcal{S}|$ is the size of the network and $D$ is the number of parameters to be estimated in the network. $N \times V/D$ means the ratio of training instances times the number of variables to the number parameters.

| Data set | $N$ | $|\mathcal{S}|$ | $D$ | Train | Valid | Test | $N \times V/D$ |
|---|---|---|---|---|---|---|---|
| NLTCS | 16 | 13,733 | 1,716 | 16,181 | 2,157 | 3,236 | 150.871 |
| MSNBC | 17 | 54,839 | 24,452 | 291,326 | 38,843 | 58,265 | 202.541 |
| KDD 2k | 64 | 48,279 | 14,292 | 180,092 | 19,907 | 34,955 | 806.457 |
| Plants | 69 | 132,959 | 58,853 | 17,412 | 2,321 | 3,482 | 20.414 |
| Audio | 100 | 739,525 | 196,103 | 15,000 | 2,000 | 3,000 | 7.649 |
| Jester | 100 | 314,013 | 180,750 | 9,000 | 1,000 | 4,116 | 4.979 |
| Netflix | 100 | 161,655 | 51,601 | 15,000 | 2,000 | 3,000 | 29.069 |
| Accidents | 111 | 204,501 | 74,804 | 12,758 | 1,700 | 2,551 | 18.931 |
| Retail | 135 | 56,931 | 22,113 | 22,041 | 2,938 | 4,408 | 134.560 |
| Pumsb-star | 163 | 140,339 | 63,173 | 12,262 | 1,635 | 2,452 | 31.638 |
| DNA | 180 | 108,021 | 52,121 | 1,600 | 400 | 1,186 | 5.526 |
| Kosarak | 190 | 203,321 | 53,204 | 33,375 | 4,450 | 6,675 | 119.187 |
| MSWeb | 294 | 68,853 | 20,346 | 29,441 | 3,270 | 5,000 | 425.423 |
| Book | 500 | 190,625 | 41,122 | 8,700 | 1,159 | 1,739 | 105.783 |
| EachMovie | 500 | 522,753 | 188,387 | 4,524 | 1,002 | 591 | 12.007 |
| WebKB | 839 | 1,439,751 | 879,893 | 2,803 | 558 | 838 | 2.673 |
| Reuters-52 | 889 | 2,210,325 | 1,453,390 | 6,532 | 1,028 | 1,540 | 3.995 |
| 20 Newsgrp | 910 | 14,561,965 | 8,295,407 | 11,293 | 3,764 | 3,764 | 1.239 |
| BBC | 1058 | 1,879,921 | 1,222,536 | 1,670 | 225 | 330 | 1.445 |
| Ad | 1556 | 4,133,421 | 1,380,676 | 2,461 | 327 | 491 | 2.774 |

Table 5: Running time of 4 algorithms on 20 data sets, measured in seconds.

| Data set | PGD | EG | SMA | CCCP |
|---|---|---|---|---|
| NLTCS | 438.35 | 718.98 | 458.99 | 206.10 |
| MSNBC | 2720.73 | 2917.72 | 8078.41 | 2008.07 |
| KDD 2k | 46388.60 | 22154.10 | 27101.50 | 29541.20 |
| Plants | 12595.60 | 10752.10 | 7604.09 | 13049.80 |
| Audio | 19647.90 | 3430.69 | 12801.70 | 14307.30 |
| Jester | 6099.44 | 6272.80 | 4082.65 | 1931.41 |
| Netflix | 29573.10 | 27931.50 | 15080.50 | 8400.20 |
| Accidents | 14266.50 | 3431.82 | 5776.00 | 20345.90 |
| Retail | 28669.50 | 7729.89 | 9866.94 | 5200.20 |
| Pumsb-star | 3115.58 | 13872.80 | 4864.72 | 2377.54 |
| DNA | 599.93 | 199.63 | 727.56 | 1380.36 |
| Kosarak | 122204.00 | 112273.00 | 49120.50 | 42809.30 |
| MSWeb | 136524.00 | 13478.10 | 65221.20 | 45132.30 |
| Book | 190398.00 | 6487.84 | 69730.50 | 23076.40 |
| EachMovie | 30071.60 | 32793.60 | 17751.10 | 60184.00 |
| WebKB | 123088.00 | 50290.90 | 44004.50 | 168142.00 |
| Reuters-52 | 13092.10 | 5438.35 | 20603.70 | 1194.31 |
| 20 Newsgrp | 151243.00 | 96025.80 | 173921.00 | 11031.80 |
| BBC | 20920.60 | 18065.00 | 36952.20 | 3440.37 |
| Ad | 12246.40 | 2183.08 | 12346.70 | 731.48 |