[Reviews · NeurIPS 2016]

Reviewer 1

Summary

The authors formulate the problem of learning parameters of a complete and decomposable SPN, based on the MLE principle, as a Signomial Programming (SP) problem. They view PGD, EG, SMA and CCCP as different levels of convex relaxation of the original SP. Projected Gradient Decent (PGD) is used in existing works ([6],[16]). Drawbacks: the projection step hurts the convergence and will often lead to solutions on the boundary of the feasible region. Sequential Monomial Approximation (SMA) and the Convex-Concave Procedure (CCCP) are two approaches for optimization the SP. SMA is general for any SP, and CCCP is claimed to exploit the structure of the specific SPs that match SPNs. EM is shown to result from CCCP, and is thus justified. Experiments compare the 4 methods (PGD, EG, SMA, CCCP) on 20 benchmark data sets. CCCP consistently converges within the smallest number of iterations to the solution with the highest likelihood compared to the other methods.

Qualitative Assessment

This is a neat work that formulates the problem in a way that allows to view how different existing techniques are optimizing the objective. Both theory and experiments identify CCCP as the most promising among the different algorithm, which is a useful result. Minor comment: “EG” acronym is presented without the full name or reference (though both are given in the supplemental material).

Confidence in this Review

1-Less confident (might not have understood significant parts)


Reviewer 2

Summary

This paper studies the weight learning problem in tractable (complete, decomposable) sum product networks (SPN). Basically, it is shown that in these cases, the SPN can be seen as a mixture of univariate distributions. From this, they show that max likelihood learning can be characterized as a certain kind of geometric program, which then leads to better empirical performance.

Qualitative Assessment

The paper's optimization formulation is well-motivated, but it would have been good to see 1) comparison to lifted learning algorithms , eg. Jan Van Haaren, Guy Van den Broeck, Wannes Meert, Jesse Davis: Lifted generative learning of Markov logic networks. Machine Learning 103(1): 27-55 (2016) 2) and a deeper discussion on arithmetic circuits. (In particular, which class of ACs do complete-decomposable SPNs correspond to?) In terms of readability, an example on what these univariate distributions look like in Section 3.1 would be useful. 3) line 153-154 on recursive merging: can you provide a formal claim

Confidence in this Review

2-Confident (read it all; understood it all reasonably well)


Reviewer 3

Summary

This paper casts SPN parameter learning into a signomial program and discusses different approximate solution algorithms to this program. It experimentally shows that the CCCP approximation converges in fewer iterations.

Qualitative Assessment

The paper appears technically sound. While the experiments show good performance, it does so by using update equations identical to [14], where their superior performance was already shown. There are some insights, by casting the problem into a signomial program, and how existing algorithms are approximations to this exact problem. On the other hand, the signomial encoding is very generic (it seems to apply to any probability distribution, which are all miltilinear function) and there is not much SPN-specific intuition. The fact that experiments are a function of the number of iteration instead of time makes me think that the time experiments are much less impressive. The presentation is fine, although still too technical and using heavy notation.

Confidence in this Review

2-Confident (read it all; understood it all reasonably well)


Reviewer 4

Summary

The paper starts by proving that all SPNs can be represented by shallow SPNs (SPNs with depth less than or equal to 3). Then it proves that this type of paper can be learn by using an EM approach (this finding is obtained by solving an optimization equation).

Qualitative Assessment

I think this paper biggest value is that it proved that using EM in SPN's learning is mathematically correct, and not just intuitively so. That being said, I think the paper has a strong resemblance to "On Theoretical Properties of Sum-Product Networks" by Robert Peharz, Sebastian Tschiatschek, Franz Pernkopf, and Pedro Domingos. I also think some sections could use a deeper explanation. For example, Theorem 3 is straightforward for depth less than or equal to 3. However, I don't think that is the case for depth greater than 3. An explanation of that case would be really useful. I really enjoyed seeing the math of the equation optimization.

Confidence in this Review

2-Confident (read it all; understood it all reasonably well)


Reviewer 5

Summary

The authors present background on SPNs and signomial programming. They show SPNs are equivalent to a mixture of a huge number of trees; this formulation lets them show that finding the MLE for the parameters of an SPN is equivalent to a signomial program which they write out. They develop two methods (SMA and CCCP) for solving this program and compare with two other parameter-learning algorithms (PGD and EG). They find that CCCP converges more quickly and learns a model with lower training-set negative-log-likelihood. They also improve the models produced by LearnSPN by fine-tuning their parameters using CCCP; after doing this the models become more competetive against ID-SPN.

Qualitative Assessment

1. I don't see a definition/name for the acronym "EG". 2. You should consider citing these papers: * "On the Robustness of Most Probable Explanations" by Chan and Darwiche, UAI 2006. It defines a "complete sub-circuit", which is very similar to "Induced SPN" from Defn. 2. * "Greedy Structure Search for Sum-Product Networks" by Dennis and Ventura, IJCAI 2015. In Theorem 1 it proves that an SPN computes the sum of all the unique trees in it, similar to the second claim in your Theorem 3. This paper is very well-written and provides an interesting analysis of SPN parameter-learning. The main take-away is that CCCP is really good for this problem. That result could be criticized by saying that CCCP is not a new learning method since it uses the same update equation as EM. Moreover, the first SPN paper by Poon remarks that gradient descent (PGD here) was slow and gave poor results and recommends hard-EM instead; this is similar in spirit to what the experiments in this paper show. However, as the authors state it is interesting to come to the same update equation using two very different analyses, and I also think the derivation and analysis of SMA and CCCP is interesting in its own right. Well done.

Confidence in this Review

3-Expert (read the paper in detail, know the area, quite certain of my opinion)


Reviewer 6

Summary

This paper derives methods to train SPNs starting from a perspective very different from methods established in literature. SPN training is viewed as a signomial program, and different approximate solutions to this problem are proposed. A theoretical and experimental analysis shows that the best method, called concave-convex procedure (CCCP), is equivalent to the already established Expectation Maximization for SPNs.

Qualitative Assessment

The single contribution of the paper which is relevant in practice is an alternative derivation of an existing method (Expectation Maximization for learning SPN weights). While this is an interesting result, I think that it does not grant alone a publication in NIPS since it's hard to imagine how this can contribute to better theoretical understanding or practical applications of SPNs. The interpretation of SPNs as mixtures of tree structured SPNs, which is reported as a novelty by the authors, was actually first derived in [Dennis and Vantura, Greedy Structure Search for Sum-Product Networks, 2015]. The paper is overall well written, clearly structured and the derivation of the results is really interesting. My main concern, as detailed above, is that in my opinion the potential impact of this paper is low, and the novelty is also somewhat limited due to the fact that the interpretation of SPN as mixture of trees was already given in [Dennis and Vantura, Greedy Structure Search for Sum-Product Networks, 2015] and that this is basically just an alternative derivation of EM. So, I think the paper is (in the current form) a bit under the threshold for NIPS. The experiment section is also not surprising, since basically it takes a SPN which is not trained, but in which weights are initialized with heuristic methods, and trains them with EM - which naturally improves a bit the log likelihood. The title "unified approach" is misleading, since not all learning methods for SPNs are covered by this approach - in particular, the hard-gradient and hardEM methods that have been used successfully in the past (see [Discriminative Learning of Sum-Product Networks, Gens and Domingos, NIPS 2012]). Small remark - in line 311: it is not always true that size is smaller in LearnSPN for 20News and AD (see line 574). Overall, an interesting paper in terms of original derivation of results which have potential for future development, but the results are not novel enough, in my opinion, to grant publication at NIPS.

Confidence in this Review

3-Expert (read the paper in detail, know the area, quite certain of my opinion)